# How Researchers, Clinicians and Patient Advocates Can Accelerate Lobular Breast Cancer Research

**DOI:** 10.3390/cancers13133094

**Published:** 2021-06-22

**Authors:** Leigh Pate, Christine Desmedt, Otto Metzger, Laurie Burgess Hutcheson, Claire Turner, Siobhán Freeney, Steffi Oesterreich

**Affiliations:** 1Lobular Breast Cancer Alliance, Breast Cancer Advocate, Seattle, WA 98117, USA; 2Laboratory for Translational Breast Cancer Research, Department of Oncology, KU Leuven, 3000 Leuven, Belgium; christine.desmedt@kuleuven.be; 3Dana-Farber Cancer Institute, Harvard Medical School, Boston, MA 02215, USA; otto_metzger@dfci.harvard.edu; 4Lobular Breast Cancer Alliance, Plymouth, MA 02360, USA; LaurieHutcheson@lobularbreastcancer.org; 5Lobular Breast Cancer UK, UK Patient Advocate Representative for European Lobular Breast Cancer Consortium, Manchester, Greater Manchester M32 8PA, UK; claire@lobularbreastcancer.org.uk; 6European Lobular Breast Cancer Consortium, Y25HP26 Dublin, Wexford, Ireland; s-freeney@hotmail.com; 7Department of Pharmacology and Chemical Biology, University of Pittsburgh, Pittsburgh, PA 15261, USA; 8Magee Women’s Research Institute, Pittsburgh, PA 15213, USA; 9UPMC Hillman Cancer Center, Pittsburgh, PA 15232, USA

**Keywords:** breast cancer, lobular, ductal, invasive lobular carcinoma, ILC, patient advocacy, advocacy, research advocacy

## Abstract

**Simple Summary:**

This commentary reflects a collaborative effort between international Invasive Lobular Carcinoma (ILC)-focused breast cancer researchers, clinicians and patient advocate leaders. It offers a perspective on the progress made in ILC research in recent years and discusses the recent rise in patient advocate involvement to advance ILC research, raise awareness and educate about this disease. It outlines several distinct challenges in conducting ILC research and describes opportunities and suggestions for ways researchers, clinicians and advocates can work together to advance ILC research to develop new therapies and refine the care offered to patients.

**Abstract:**

Breast cancer research and therapies have significantly advanced in recent years. However, Invasive Lobular Carcinoma (ILC), the second most common histological type of breast cancer and the sixth most frequently diagnosed cancer of women, has not always benefited from critical analysis, missing opportunities to better understand this important subtype. Recent progress understanding the biological and behavioral differences of ILC demonstrates that it is a unique subtype of breast cancer which can respond differently to common therapies. These new insights have increased interest in researching lobular breast disease. Concurrently, the formation of motivated patient-led advocacy organizations working in partnership with basic, translational and clinical researchers creates new opportunities, including connecting a dispersed patient population to research, encouraging new research funding and connecting patient advocates to researchers to advance common goals. This commentary will explore the unprecedented opportunity to drive multidisciplinary, multicenter and international collaborative research into lobular breast cancer that builds on recent research progress. Collaborative research partnerships that include advocates can result in a better understanding of ILC, identify targeted therapies and refine standard of care therapies that are currently equally applied to all breast cancers, resulting in improvements in the diagnosis, treatment and follow-up care for patients with ILC.

## 1. Historic View of Lobular Carcinoma Is Challenged by New Evidence

Invasive Lobular Carcinoma (ILC) is the second most frequently diagnosed histological type of breast cancer. Invasive Ductal Carcinoma (IDC), formally called “invasive breast carcinoma of no special type”, is the most common histological type of breast cancer. ILC accounts for 10–15% of all breast cancers, and alone is the sixth most frequently diagnosed cancer of women in the US [1]. Incidence rates are expected to be similar in Europe; for example, in Belgium, ILC accounts for 14% of all breast cancers diagnosed in 2014–2018 (Belgian Cancer Registry, Brussels, Belgium). In 2021, breast cancer became the most frequently diagnosed cancer in the world, making ILC a prevalent and growing public health concern worldwide. 

Lobular breast cancer was first described a century ago by James Ewing [2], and the first detailed characterization was provided by Frank W Foote and Fred W Stewart [3]. Although the article focused on early lesions such as lobular carcinoma in situ (LCIS), it also described the invasive lobular counterpart and recognized lobular features such as “rare mitoses” and “loss of cohesion”. While progress was made in subsequent years, including defining subtypes within ILC in the 1970s [4], the prevailing view was that ILC was simply another estrogen receptor (ER)-positive breast cancer. At the time, this view was shared by most, including the eminent breast cancer researchers and physicians Edwin R. Fisher and Bernard Fischer, who concluded, “*Since the overall 5 and 10-year survival rates of patients with infiltrating lobular carcinoma are reputed to be similar … there are no special considerations regarding its management. They should be similarly treated.*” Furthermore, “*…whatever information is obtained in the next few years from on-going clinical trials regarding management of breast cancer will be directly applicable to this variant*” [5].

This 40+ year old historic view has been challenged recently, opening doors to understand the biology of this disease and refine treatment strategies that can improve outcomes for patients.

The historic view that ILC and IDC survival rates are equivalent, and the perception by many that ILC is a less lethal type of breast cancer, have been challenged by more recent retrospective analyses suggesting that while short-term survival rates for lobular are similar to ductal cancers, long-term survival for patients with ILC may be worse [6,7,8]. This is particularly surprising given ILC typically has better prognostic factors such as high ER, progesterone receptor (PR) and low proliferation marker Ki67 [9,10,11,12].

Recent studies have shed new light on ILC’s different biology, clinical presentation, disease behavior, and metastasis [11,13,14]. Findings include differences in enriched molecular features in lobular tumors that could lead to future targeted therapies [15,16,17,18,19].

Evidence mounts that lobular tumors may respond differently to commonly applied therapies including adjuvant (post-operative) chemotherapies [20,21] and endocrine therapy. For example, in a retrospective analysis of the phase III study BIG 1-98, post-menopausal women diagnosed with ILC had worse survival outcomes when treated with tamoxifen compared to letrozole [22]. These results suggested that the efficacy of commonly administered endocrine treatments could differ between patients with IDC and ILC.

Lobular breast cancers have long been known to be more occult on imaging in the primary and metastatic setting. Progress has been made with [18F]-fluoro-estradiol (FES) positron-emission tomography (PET) imaging [23,24] and Fluciclovine (18F-FACBC) PET [25] and additional trials are ongoing, including one focused on FES-PET/CT Imaging of ILC (NCT04252859, https://clinicaltrials.gov/ct2/show/NCT04252859, accessed on 6 May 2021).

In the current era of research that emphasizes de-escalation of treatment, the assays and prognostic tests used universally were developed primarily on IDC. Given increasing reliance on these tests for clinical decisions, validating their efficacy in lobular disease is important [26,27]. Additionally, a potentially clinically relevant prognostic signature called LobSig [28] is in development.

Recent studies and clinical trials such as PELOPS (NCT02764541, https://clinicaltrials.gov/ct2/show/NCT02764541, accessed on 6 May 2021), ROLO (NCT03620643, https://clinicaltrials.gov/ct2/show/NCT03620643, accessed on 6 May 2021), GELATO (NCT03147040, https://clinicaltrials.gov/ct2/show/NCT03147040, accessed on 6 May 2021), ROSALINE (NCT04551495, https://clinicaltrials.gov/ct2/show/NCT04551495, accessed on 6 May 2021), TBCRC037 (NCT02206984, https://clinicaltrials.gov/ct2/show/NCT02206984, accessed on 6 May 2021), and more have begun to explore alternate or refined approaches that can lead to better outcomes. However, there is much more research needed to better understand how to refine treatments for patients with ILC.

## 2. First International Lobular Breast Cancer Symposium Sparks New Patient Advocacy

Historically, patients with ILC were a predominantly invisible and silent presence within the breast cancer research and patient support environment. This began to change in September 2016, when the 1st International Lobular Breast Cancer Symposium held in Pittsburgh, Pennsylvania welcomed thirty patient advocates from around the United States to the first ever meeting dedicated to ILC research, alongside more than 100 laboratory-based and clinical researchers.

The 1st ILC Symposium was hosted by UPMC Hillman Cancer Center and chaired by Steffi Oesterreich, PhD, Nancy Davidson, MD and Otto Metzger, MD with the support of the NCI, the Breast Cancer Research Foundation, Susan G Komen and other foundations and convened patient advocates under the leadership of Heather Hillier, at the time a member of the Pittsburgh patient advocacy group bcRAN [29,30]. The symposium featured events organized for and by patient advocates, including an evening educational Q&A. Casual breakfast table discussions with experts in basic, translational and clinical research joined by 5–6 advocates sparked lively questions and ideas. A break-out meeting for patient advocates led to an informal discussion of patient needs and raised ideas for how patients can advance lobular research.

These events were not only successful at the time, but they emphasized the mutual dependency between ILC researchers and patient advocates. Discussions highlighted several opportunities to collaborate and to build a strong foundation for progress:Patient advocates were hungry for quality information about their disease which was often omitted in breast cancer educational resources. This left patients without the tools to be informed partners in their self-care, lacking even basic information such as unusual sites where lobular tumors can spread and symptoms of recurrence to report to their treating physicians.Patient advocates expressed frustration and lack of confidence in standard of care imaging and therapies that were primarily designed and trialed on patients with IDC, but were not always the best fit or the efficacy was unknown in patients with ILC.Advocates were eager to identify and bridge gaps between research and patients and support studies and trials focused on ILC.Researchers and clinicians found that they needed critical input from advocates to identify the most important needs for patients.Researchers and clinicians needed the patient voice to advocate with funders, institutions and organizations to make the case for more ILC-focused research and to raise awareness of this disease in the broader breast cancer and research community.

## 3. Role of Patient Advocacy in Lobular Breast Cancer

As with other advocacies, lobular patient advocacy can take many forms. Advocates can raise awareness of ILC and educate within the broader breast cancer community and the public. They can also serve as peer mentors to provide support to other patients. Advocates can fundraise to support both lobular advocacy organizations and to directly fund research. Advocates can lobby to change government policies and support research funding.

Patients with ILC (or their friends, families and others) can also serve as research advocates. Research advocates act as a liaison between researchers and patients and work directly with researchers to inform the research questions that need to be answered to meet real patient needs. Research advocates can also inform the design of studies and trials and translate and communicate research results back to patient communities.

Specifically, research advocates can partner with researchers to shape research proposals and grant applications, review lay summaries, confirm their interest through providing letters of support, participate in consumer grant reviews, help design clinical trial protocols, identify research priorities, work within cancer research institutions through local patient advocacy groups or as part of the Institutional Review Board (IRB), attend professional conferences and events, conduct and present patient-driven research, and share the results of research in the broader breast cancer community.

## 4. Launching of the Lobular Breast Cancer Alliance and Global ILC-Focused Advocacy Efforts

Patient advocates attending the 1st ILC Symposium, led by Leigh Pate and working with symposium co-chair Steffi Oesterreich PhD, developed a white paper [31] conveying the advocate perspective that ILC was a distinct disease that requires refinements in screening, patient care, patient and provider education and substantive measures to accelerate research. The conclusions of this white paper became the basis of the mission and goals driving the US-based advocate-driven organization Lobular Breast Cancer Alliance (LBCA, lobularbreastcancer.org) which formally launched in December 2017.

Today, LBCA is the largest lobular advocacy organization with a global audience, a comprehensive website and an international Scientific Advisory Board. LBCA’s mission is “to make all who are touched by ILC aware of its unique characteristics and the critical need for more ILC research; to be the go-to source for information on ILC studies, clinical trials and educational tools; to foster partnerships among patients, scientists, clinicians and organizations to increase dialogue about ILC and research advocacy; and to fund vital ILC research.” LBCA engages breast cancer researchers, clinicians, patients, advocates, organizations and the public through online communications, strategic partnerships, events and research collaborations with the goal of advancing ILC research and educating about lobular breast disease.

The launch of LBCA in the United States inspired additional local and regional lobular breast cancer advocacy in the US such as the Dynami Foundation (www.dynamifoundation.org, accessed on 6 May 2021) and ignited interest in additional ILC patient advocacy globally. Aspiring patient advocates around the world reached out to LBCA, wanting to join the effort and requesting assistance to organize. Independent advocates from Canada, Australia, Nigeria, across Europe, the UK, India and New Zealand are engaging with LBCA’s advocacy and communications programs, many currently without a formal lobular patient advocacy organization within their borders.

The European Lobular Breast Cancer Consortium (ELBCC) (www.elbcc.org, accessed on 6 May 2021), a research consortium of basic, translational and clinical researchers working with patient advocates from across Europe, was established in November 2018 and is sponsored by the European Cooperation in Science and Technology (COST) action LOBSTERPOT (CA19138; http://www.elbcc.org/lobsterpot.html, accessed on 17 June 2021). ELBCC integrates individual and organizational patient advocates in its efforts to create awareness for lobular breast cancer as a cancer type, advance research into ILC and improve outcomes for patients. ELBCC has defined task forces to induce multidisciplinary approaches to better understand ILC biology and tackle the key problems in ILC diagnosis and treatment. Patient advocacy, disease awareness and education, and dissemination of scientific progress and clinical advancements are among the key goals for the consortium. Within ELBCC, country-specific patient- driven advocacy efforts have been launched in Europe.

**Lobular Breast Cancer UK (LBCUK)** (Lobularbreastcancer.org.uk) is a patient-advocate-driven organization in the United Kingdom that has charitable status and a website. LBCUK’s goals are to drive education, research, policy change and ensure patient support is available for patients with ILC within outside organizations and support services. LBCUK will focus on informing and supporting patients to be their own best advocates, partnering with researchers, representing ILC advocacy at conferences and developing and driving funded lobular research programs. LBCUK has a Scientific Advisory Group involving researchers and clinicians from institutions across the UK working in partnership with the charity who provide scientific and medical advice and guidance and the opportunity to develop longer-term ILC-focused research programs.**Lobular Ireland** (LobularIreland.com) is a network of ILC advocates and breast cancer researchers from the Royal College of Surgeons, University College Dublin with a clinical advisor from Cancer Trials Ireland. Collectively, they advocate for more research into ILC and raise awareness about ILC. Lobular Ireland’s goals are to partner with researchers, build patient awareness and have ILC representation at key breast cancer conferences and seminars.

LBCA, ELBCC advocates, LBCUK, Lobular Ireland and Dynami are working to build a collaborative international lobular patient advocacy effort to accomplish shared goals (Figure 1).

In addition to these organized groups, there are a number of informal lobular patient support and communication networks with global reach. Patients with ILC have organized several support and communications forums on Facebook and other social media and organizationally affiliated patient-focused platforms that reach thousands of patients and others interested in ILC around the world, some with an international membership and some that are language- or country-focused. These forums create opportunities to share information on specific lobular clinical trials and studies broadly, activate a global advocate base and educate patients with ILC about their disease.

There are indications that the new lobular patient research advocacy is having results. Since LBCA launched in 2017, there has been a noticeable increase in clinical trials focused on ILC and an increased presence of studies at conferences and meetings, including the San Antonio Breast Cancer Symposium. New advocate-driven fundraising has initiated ILC-specific research grants, including new investigator-focused collaborative grants sponsored by LBCA with the American Society of Clinical Oncology (ASCO) and American Association for Cancer Research (AACR), as well as locally driven funding for US-based institutional and regional lobular research programs such as the Dynami Foundation. Globally, the dialogue between clinicians, researchers and patient advocates is growing, assisted by the increasing access of patient advocates to register for large conferences for free and increasing use of virtual technology in meetings, which eliminates travel expenses.

## 5. Elements of Successful Researcher and Patient Research Advocate Collaboration

There are many examples of successful partnerships of patient research advocates and researchers across medicine. The Specialized Programs of Research Excellence (SPORE) within the Translational Research Program of the National Cancer Institute (https://trp.cancer.gov/, accessed on 6 May 2021) closely integrates advocates into many research activities. GRASP (Guiding Researchers and Advocates to Scientific Partnerships) (https://graspcancer.org/about/, accessed on 6 May 2021) pairs advocates with scientists to review and discuss scientific posters at leading breast cancer conferences. ROS1ders (https://ros1cancer.com/, accessed on 6 May 2021) focus on advancing targeted research for ROS1-positive cancers. Additionally, TBCRC (https://www.tbcrc.org/, accessed on 6 May 2021) and SWOG (https://www.swog.org/, accessed on 6 May 2021) are cancer research networks that integrate patient research advocates to serve as the voice of the patient in the development and execution of trials. In the UK, one patient advocate group (independentcancerpatientsvoice.org.uk) partners with clinicians and healthcare professionals to improve clinical research in breast cancer.

Several elements enabled the successful launch of LBCA and led to global lobular patient advocacy.

**Timing and opportunity:** As evidence mounted that ILC was a different disease, the research community realized there were unique aspects of ILC that deserved and required more research which would benefit from advocate participation. The new interest in research coincided with early communication and organization among patients with lobular breast cancer on early social media platforms, leading to strong advocate attendance at the 1st ILC Symposium and motivated engagement after the conference.**Leadership:** The prior professional experience of LBCA’s founding leader Leigh Pate in successful large-scale political and issue advocacy campaigns provided a strong foundation in organizing, strategic communications and management of volunteers and coalitions. Scientific Advisory Board founder and past-chair Steffi Oesterreich PhD leveraged her experience through various foundations and institutions, and passionately partnered with advocates to guide LBCA’s interactions within the breast cancer research community. This advocate/researcher partnership opened doors, lent professionalism and credibility to LBCA’s public content and efforts, and led to the engagement of a strong network of committed scientific advisors from diverse institutions as well as broad breast cancer organizational support.**Partnerships:** Importantly, LBCA’s leaders developed a strong working relationship and a level of trust that allowed them to pursue an ambitious agenda. LBCA’s early organizational structure was never dependent on one individual to carry the efforts, was consensus based and included a strong co-coordinator and advocate leader in Lori Petitti and a volunteer advocate steering committee with diverse backgrounds and skills.

Important elements that sustained and built ongoing relationships between lobular advocates and researchers include:**Collaborations:** A founding tenet of lobular advocate leaders internationally is grounded in collaboration—the belief that patient advocates, clinicians and researchers must work together to overcome challenges in lobular research, create change and advance patient-centered research initiatives.**Commitment to scientific review:** LBCA developed educational content on lobular breast cancer, working with Scientific Advisors who lent their expertise, making LBCA a reliable resource within the breast cancer community and to our community partners.**Dedicated communications platform:** LBCA’s website and communications platforms, as well as a growing network of advocates worldwide, provide a center of information on ILC. This benefits not only patients looking for education and resources, but researchers who have a platform to share information with an audience specifically interested in ILC.

## 6. Challenges and Opportunities for Researchers, Clinicians and Patient Advocates to Accelerate ILC Research

The rapidly expanding reach of lobular advocacy organizations creates new opportunities to bridge the traditional gaps that have challenged ILC research and give clinicians and patients the tools and confidence to make informed decisions about the best therapies to most successfully treat and manage ILC. Key challenges and opportunities are described below and schematically summarized in Figure 2.

### 6.1. Challenge #1: Define ILC Research Priorities

#### Opportunities

Conduct a global survey of clinicians, researchers and patient advocates to identify and develop agreement on ILC research priorities.

Research advocates can partner within study and trial design early in the process so research reflects patient needs and is designed from the beginning with a perspective that can make a study more impactful and successful. National lobular breast cancer advocacy organizations can provide advocates to support research initiatives, should local research institution advocacy groups lack sufficient ILC patient representation.

Conference and event organizers, cancer institutions and organizations, grant making entities and grant review committees can integrate ILC advocates into their planning and participation, particularly when ILC-relevant research questions are addressed.

Advocates with ILC can attend important meetings such as the San Antonio Breast Cancer Symposium to represent the patient voice and facilitate researcher/advocate collaborations.

Researchers and clinicians can work with their respective institutions to assure financial support for advocate attendance of conferences and meetings. Incorporating advocate scholarships as a line item into institutions’ budgets or within grants, recruiting dedicated philanthropic support for advocacy, discounting or eliminating conference registration fees for advocates and making conferences accessible online are all ways to remove barriers for advocate participation.

### 6.2. Challenge #2: Design Clinical Trials Focused on ILC and Share Trial Participation Opportunities

#### Opportunities

Conduct clinical research focused on identifying and developing appropriate ILC treatment guidelines that refine therapies for patients.

Integrate ILC into appropriate clinical trials, collect histological subtype information, and report results in a subtype-dependent manner to identify signals of responses for therapies that may have efficacy for ILC.

Lobular advocacy organizations can utilize their communications platforms and outreach capacity to raise awareness and educate about the urgent patient need for more clinical research on ILC.

Researchers can discuss innovative approaches to trial design to evaluate ILC in larger clinical research studies. For example, the PELOPS trial (NCT02764541, https://clinicaltrials.gov/ct2/show/NCT02764541, accessed on 6 May 2021) was enriched for breast cancer patients with ILC.

Health providers and advocates can press research institutions and medical professional organizations to prioritize and advance research that informs future ILC treatment guidelines.

Advocates can bridge communication gaps between researchers and dispersed patient populations by sharing information about enrolling ILC clinical trials that patients can discuss with their providers to boost enrollment.

### 6.3. Challenge #3: Standardize Diagnosis of ILC

#### Opportunities

Comprehensively evaluate current practices for diagnosis of ILC through communication with breast pathologists, which could then be used to issue standardized recommendations.

Advocates can emphasize the need for central pathology standards aimed at defining breast cancer subtypes in prospective trials and prioritize and communicate support of research and trials to improve and refine the diagnosis of ILC.

Pathologists need to be integrated into communication about trials and studies along with ILC clinicians, researchers, and advocates.

### 6.4. Challenge #4: Collect Data and Samples from Patients with ILC

#### Opportunities

More emphasis on tissue collection from patients with lobular disease is critical because not only is ILC less common than IDC, but the tremendous heterogeneity within ILC introduces additional challenges and the need for larger numbers of tissues.

Collection of fresh tissue is important to establish more ILC cell line and in vivo models which remain scarce, providing an added challenge to developing novel targeted treatments. 

Advocates can raise awareness of the need to collect and comprehensively analyze ILC tissues for better understanding of the heterogenous make-up of ILC.

Advocacy organizations can assist in future collaborative efforts to build ILC registries or tissue banks by sharing opportunities to participate in research with patients.

### 6.5. Challenge #5: Assemble and Communicate Up-to-Date and Accurate Educational and Research Information on ILC

#### Opportunities

Advocacy organizations offer growing communications platforms that make lobular research findings and educational information available to a wider audience, including patients and the public, the clinical and research community and breast cancer institutions and organizations.

Researchers and advocacy organizations can partner to disseminate new ILC-specific research findings and provide forums that promote a dialogue about research findings among researchers, clinicians and advocates.

Researchers can partner with advocacy organizations by sharing their research and providing easier-to-understand and communicate lay summaries to ease distribution.

Researchers and clinicians can assist advocacy organizations to develop educational webinars, videos, website content and presentations.

### 6.6. Challenge #6: Integrate ILC into Cancer Conferences, Professional Education and Cancer Organizations

#### Opportunities

Organizers of regional, national and international breast cancer meetings and symposia can include ILC into the program such as the ILC-specific Mini-Symposium at SABCS 2017.

Include greater emphasis of ILC, its differences and latest research in early and continuing oncology medical education. Include ILC in medical education for primary care, gynecology, GI specialists, survivorship care and nursing so key first-line providers have basic knowledge about ILC, including presentations in the breast and sites and symptoms of metastasis. Currently, patients frequently turn to outside sources such as the internet and social media groups for information about ILC and their care. In a survey of 950 LBCA website users, respondents (95% current or former patients with ILC) ranked their doctor fourth when asked for their top three sources of information about ILC [32].

Institutions can support interaction between advocates and trainees so that the next generation of breast cancer researchers appreciate ILC as a unique disease.

Advocates, researchers and clinicians can build partnerships with larger cancer organizations to integrate ILC into existing programming and funding priorities.

### 6.7. Challenge #7: Establish a Collaborative, Coordinated Worldwide ILC Research Strategy

#### Opportunities

Large ILC clinical trials and studies will likely require multicenter collaborations working with the support of patient advocacy organizations to accrue patients. Increased online communications make future multicenter and international collaborations simpler and more accessible to all.

Coordinated research efforts can help resolve inconsistencies in tissue and data collection that is not standardized across institutions, which can make it difficult to compare between multiple institutions and studies.

Launching trials and studies in a coordinated manner can create opportunities to answer multiple questions and leverage opportunities to utilize limited resources for maximum benefit.

Lobular patient advocacy groups can be supportive agents, partners and participants in global collaborative efforts.

### 6.8. Challenge #8: Encourage Funding Focused on ILC Research

#### Opportunities

Advocates and lobular organizations can and are fundraising to create new lobular research-specific funding sources.

Encourage cancer research institutions and traditional research funders to institute guidance encouraging proportional allocation of funding and resources to different molecular and histological types of breast cancer in their representation and focus (i.e., 10–15% to lobular).

Advocates can encourage and support more basic science and preclinical work in ILC, emphasizing fundamental questions needing to be addressed, including a better understanding of the role of E-cadherin. Generating models representing heterogenous forms of ILC (including primary classic ILC) is critical.

Advocates and advocacy organizations can partner with researchers on developing grant and funding requests, lending the patient perspective as well as the additional assets an organization with communication resources can offer to strengthen proposals.

Advocates can participate in formal grant reviews.

## 7. Discussion

### How Researchers, Clinicians and Advocates Can Work Together to Move Lobular Research Forward in the Future

Developing a multicenter lobular breast cancer research strategy with a partnership and infrastructure that is collaborative and internationally cohesive can overcome some of the challenges for ILC research. Collaborative efforts are particularly important to conduct larger trials and studies, and to coordinate research efforts to maximize impact. It can also provide access to shared resources such as models, tissues and data. Collaboration provides a framework so that lobular research efforts and scarce resources are not duplicative and are working efficiently towards common patient-prioritized goals.

A future collaboration could incorporate institutions researching ILC and patient advocacy organizations. It could identify shared research priorities of scientists and patient advocates, so research aligns with patient priorities. It could facilitate regular meetings to exchange ideas, share research findings, convene working groups and mentor trainees. It can be a resource for training and education in lobular breast cancer. Potentially, this entity can grow to serve as a center for coordinated lobular research. Most importantly a collaborative, multicenter international effort can share a single focus of advancing lobular breast cancer research, filling a void that has left lobular breast cancer without a focus and a voice for the last 100 years.

Together, we need to work out how to integrate ILC into conferences and meetings and into the objectives of appropriate studies—this is required to better understand the differences between the different histological subtypes, and to identify the best paths forward. Ultimately, patients can have the information they need to make informed decisions about their care, and clinicians can have the informed treatment guidelines and resources to give their patients the best therapies.

## 8. Conclusions

Recent discoveries in understanding ILC’s unique biology and behavior have renewed interest in understanding its differences and how that might influence commonly applied therapies. This reinvigorated interest, combined with the rise and rapid growth of international lobular patient advocacy organizations, can drive progress in developing the therapies that will benefit patients.

Moving forward with coordinated, collaborative research that integrates basic, translational and clinical researchers with a strong patient advocate presence can overcome the research challenges of lobular breast cancer and improve the diagnosis, treatment and follow-up care for patients.

## Figures and Tables

**Figure 1 cancers-13-03094-f001:**
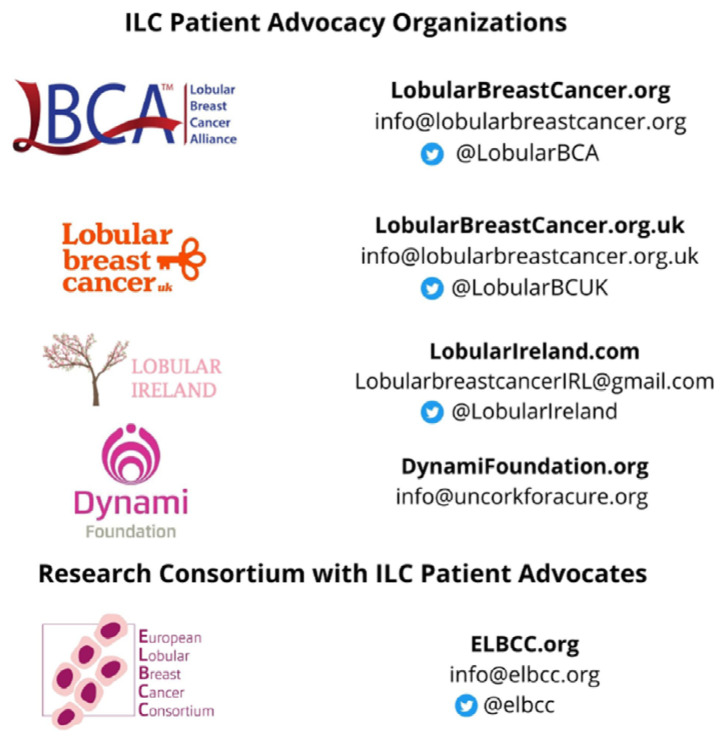
Lobular Breast Cancer Patient Advocacy Entities. Researchers, clinicians and advocates interested in getting involved in organizational activities or lobular patient advocacy can contact these organizations to learn more.

**Figure 2 cancers-13-03094-f002:**
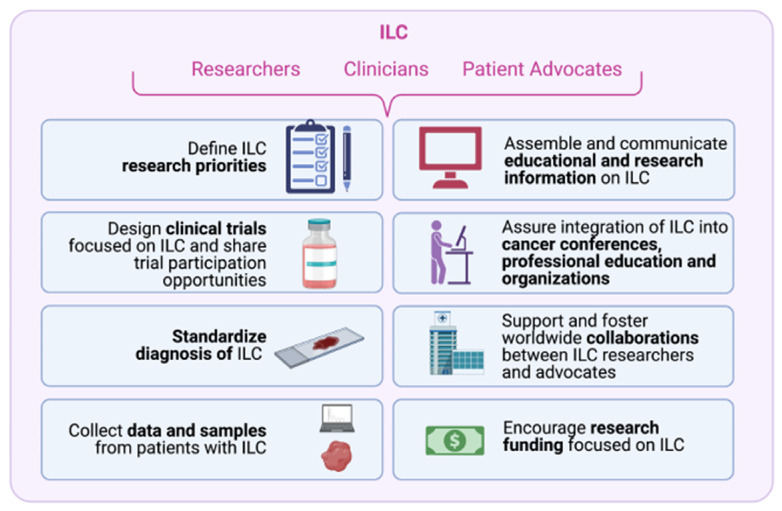
Challenges and opportunities in ILC research. Working together, researchers and clinicians can make progress to overcome challenges and advance research into ILC.

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
