# Peer review of "How Researchers, Clinicians and Patient Advocates Can Accelerate Lobular Breast Cancer Research"

_cancers, 2021, doi:10.3390/cancers13133094_

Round 1
Reviewer 1 Report
This a well written and important article highlighting the important role of patient advocates in general, and how they can be instrumental in raising awareness and driving research in rarer special subtypes of cancer such as ILC.
Some minor comments:
- The formatting of the manuscript needs some attention: there are many examples of words running into each other without any spaces and the line spacing changes in different paragraphs.
- Bottom of page 1: where the authors state that ILC “is the sixth most frequently diagnosed cancer of women in the US”. Please can the authors also mention the frequency in Europe and that there is an increasing incidence of ER+ breast cancer including ILC in developing countries due to changes in lifestyle, indicating this is a worldwide issue.
- Under challenge 4 it would be useful to also mention that ILC cell lines and in vivo models are scarce, which is an added challenge to developing novel targeted treatments. Collection of fresh tissue will be important to establish more ILC models.
Author Response
We appreciate the positive feedback from the reviewers.
Below please find our responses to the concerns raised:
- The formatting of the manuscript needs some attention.
Response: We apologize for these issues, and have carefully edited the revised manuscript.
- Bottom of page 1: where the authors state that ILC “is the sixth most frequently diagnosed cancer of women in the US”. Please can the authors also mention the frequency in Europe and that there is an increasing incidence of ER+ breast cancer including ILC in developing countries due to changes in lifestyle, indicating this is a worldwide issue.
Response: These are excellent points, especially given large audience of Cancers in Europe and the increasing diagnosis of breast cancers worldwide. In 2021 breast cancer became the world’s most prevalent cancer. It turns out to be a challenge to find accurate numbers of the different histological breast cancer subtypes for Europe or globally. We have therefore decided to include a number for Belgium (the country of co-author Dr Christine Desmedt), and also commented on increasing incidence. The new text states :
"Invasive Ductal Carcinoma (IDC), formally called “invasive breast carcinoma of no special type”, is the most common histological type of breast cancer. ILC accounts for 10-15% of all breast cancers, and alone is the sixth most frequently diagnosed cancer of women in the US 1. Incidence rates are expected to be similar In Europe, for example, in Belgium ILC accounts for 14% of all breast cancers diagnosed 2014-2018 (Belgian Cancer Registry, Brussels). In 2021, breast cancer became the most frequently diagnosed cancer in the world, making ILC a prevalent and growing public health concern worldwide."
- Under challenge 4 it would be useful to also mention that ILC cell lines and in vivo models are scarce, which is an added challenge to developing novel targeted treatments. Collection of fresh tissue will be important to establish more ILC models.
Response: This is also an excellent suggestion, and we included language towards this point. Specifically, we added under Challenge 4: “Collection of fresh tissue is important to establish more ILC cell line and in vivo models which remain scarce, providing an added challenge to developing novel targeted treatments.”
Reviewer 2 Report
I thank the authors for preparing this important and relevant commentary, and I thank the editors for the opportunity to review it. The messages are clear and presented in a well-structured manner and I think the commentary is fit for publication. If I can offer a suggestion, the authors could consider going deeper into the connection between actual types of research and the role of researchers, clinicians and patient advocates. I'm not only thinking of clinical studies, but also about the importance of preclinical work and fundamental science, the utility of in vivo vs in vitro vs in silico or other models, and some of the current challenges in ILC research (e.g. a lack of reliable and relevant cell line models representing primary classic ILC, and successes of models of ILC, as well as remaining challenges). The authors may decide whether they could address these topics while keeping the text succinct and focused.
Author Response
I thank the authors for preparing this important and relevant commentary, and I thank the editors for the opportunity to review it. The messages are clear and presented in a well-structured manner and I think the commentary is fit for publication. If I can offer a suggestion, the authors could consider going deeper into the connection between actual types of research and the role of researchers, clinicians and patient advocates. I'm not only thinking of clinical studies, but also about the importance of preclinical work and fundamental science, the utility of in vivo vs in vitro vs in silico or other models, and some of the current challenges in ILC research (e.g. a lack of reliable and relevant cell line models representing primary classic ILC, and successes of models of ILC, as well as remaining challenges). The authors may decide whether they could address these topics while keeping the text succinct and focused.
Response: This is a great suggestion, thank you. We originally tried to refrain from including specific research topics, as there are so many and it is difficult to choose some in an unbiased way. However, Reviewer 1 has also mentioned the need to generate more ILC cell line and in vivo models, due to the scarcity of such models. We have therefore included a comment towards this research focus under Challenge #4. In addition, we also included a sentence on the importance of basic science and preclinical work in ILC, to address fundamental questions, including the role of E-cadherin, and the generation of models representing primary classic ILC.
Specifically under challenge #8, we added: “Advocates can encourage and support more basic science and preclinical work in ILC, emphasizing fundamental questions needing to be addressed, including a better understanding of the role of E-cadherin. Generating models representing heterogenous forms of ILC (including primary classic ILC) is critical.”
Reviewer 3 Report
Although this commentary may be interesting and potentially publishable in some lower-impact journal, I do not see it having enough priority to be published in this journal.
Author Response
Although this commentary may be interesting and potentially publishable in some lower-impact journal, I do not see it having enough priority to be published in this journal.
Response: We appreciate the fact that this indeed is not the “typical” scientific paper published in Cancers. However, we strongly believe that this Commentary is a perfect fit for the journal since 1) it will be part of a special issue on lobular cancer, covering all aspect of the disease including how we can increase research effort on this traditionally understudied subtype of breast cancer, and 2) we feel it is critical for the scientific audience to learn and appreciate the role of advocates in cancer research, and specifically how advocates can work hand in hand with researchers and clinicians – the topic of this commentary.
Round 2
Reviewer 3 Report
The author responded to the queries. The paper is publishable.